# Evaluation and deployment of isotype-specific salivary antibody assays for detecting previous SARS-CoV-2 infection in children and adults

Amy C. Thomas [1,2✉], Elizabeth Oliver [2], Holly E. Baum[2], Kapil Gupta [3,4,5], Kathryn L. Shelley [3,4,6], Anna E. Long[7], Hayley E. Jones[1], Joyce Smith [2], Benjamin Hitchings [2], Natalie di Bartolo[3,8], Kate Vasileiou [3], Fruzsina Rabi[3], Hanin Alamir[2], Malak Eghleilib[2], Ore Francis [9], Jennifer Oliver[10], Begonia Morales-Aza[2], Ulrike Obst[11], Debbie Shattock[12], Rachael Barr[2,13], Lucy Collingwood[10], Kaltun Duale [2], Niall Grace[10], Guillaume Gonnage Livera[10], Lindsay Bishop[10], Harriet Downing[14], Fernanda Rodrigues[15,16], Nicholas Timpson [1,17], Caroline L. Relton [17], Ashley Toye [3,8], Derek N. Woolfson [3,4,6], Imre Berger [18], Anu Goenka[2,19], Andrew D. Davidson[11], Kathleen M. Gillespie[7], Alistair J. K. Williams[7], Mick Bailey[9], Ellen Brooks-Pollock [1], Adam Finn[1,2,19,23], Alice Halliday[2,23] & the CoMMinS Study Team*

## Abstract

**Background** Saliva is easily obtainable non-invasively and potentially suitable for detecting both current and previous SARS-CoV-2 infection, but there is limited evidence on the utility of salivary antibody testing for community surveillance.

**Methods** We established 6 ELISAs detecting IgA and IgG antibodies to whole SARS-CoV-2 spike protein, to its receptor binding domain region and to nucleocapsid protein in saliva. We evaluated diagnostic performance, and using paired saliva and serum samples, correlated mucosal and systemic antibody responses. The best-performing assays were field-tested in 20 household outbreaks.

**Results** We demonstrate in test accuracy ($N = 320$), spike IgG (ROC AUC: 95.0%, 92.8–97.3%) and spike IgA (ROC AUC: 89.9%, 86.5–93.2%) assays to discriminate best between pre-pandemic and post COVID-19 saliva samples. Specificity was 100% in younger age groups (0–19 years) for spike IgA and IgG. However, sensitivity was low for the best-performing assay (spike IgG: 50.6%, 39.8–61.4%). Using machine learning, diagnostic performance was improved when a combination of tests was used. As expected, salivary IgA was poorly correlated with serum, indicating an oral mucosal response whereas salivary IgG responses were predictive of those in serum. When deployed to household outbreaks, antibody responses were heterogeneous but remained a reliable indicator of recent infection. Intriguingly, unvaccinated children without confirmed infection showed evidence of exposure almost exclusively through specific IgA responses.

**Conclusions** Through robust standardisation, evaluation and field-testing, this work provides a platform for further studies investigating SARS-CoV-2 transmission and mucosal immunity with the potential for expanding salivo-surveillance to other respiratory infections in hard-to-reach settings.

### Plain Language Summary

If a person has been previously infected with SARS-CoV-2 they will produce specific proteins, called antibodies. These are present in the saliva and blood. Saliva is easier to obtain than blood, so we developed and evaluated six tests that detect SARS-CoV-2 antibodies in saliva in children and adults. Some tests detected antibodies to a particular protein made by SARS-CoV-2 called the spike protein, and these tests worked best. The most accurate results were obtained by using a combination of tests. Similar tests could also be developed to detect other respiratory infections which will enable easier identification of infected individuals.

A full list of author affiliations appears at the end of the paper.

Antibody detection has proven critical for conducting epidemiological surveillance, investigating the natural history of SARS-CoV-2 and assessing novel vaccine candidates throughout the ongoing COVID-19 pandemic[1–3]. Serological studies have demonstrated antibodies to be correlates of protection against (re)infection[4], with antibodies specific to spike protein and its receptor binding domain (RBD) region demonstrated to neutralise viral binding and entry[5]. Currently widely used COVID-19 vaccines generate immune responses to the spike protein and serological studies have been central to vaccine evaluation. A corollary is that antibody responses to the nucleocapsid (N-protein) offer a means to differentiate infected from vaccinated individuals in settings where the vaccines utilised include only the spike antigen.

While it has been shown that antibodies to SARS-CoV-2 can be measured in saliva[6–8], there has been limited evaluation of the suitability and utility of salivary immunoassays for detecting recent infection in populations of children and adults, particularly in more recent months when new variants have been circulating and vaccination coverage is high in many countries[9]. In saliva, secretory IgA (sIgA) and IgG are the principal antibody classes: IgA is mostly produced by local, mucosal plasma cells while IgG is mostly derived from the blood by passive diffusion, mainly across gingival crevicular epithelium[10,11]. The tropism of SARS-CoV-2 for cells in the respiratory tract suggests that consequent local generation of mucosal IgA antibodies may play an important role in protection and limiting onward transmission, while salivary IgG may be a proxy for systemic immunity[12–15].

As SARS-CoV-2 transitions to endemicity, monitoring infection, individual and population immunity and re-infection through antibody responses will be important for mitigating against future outbreaks, with success, in part, dependent on robust, well-characterised assays which can be used on easily obtained samples. Despite this, large-scale epidemiological studies using mucosal saliva samples are uncommon. Challenges exist in collection and handling of specimens to prevent degradation by sample proteases, as well as in individual variation in salivary production and composition[10]. Furthermore, mucosal immunoassays can suffer from increased assay background due to non-specific, high avidity binding of multimeric immunoglobulin[16] for low-affinity antigen, reducing discriminatory power in diagnostic tests and reproducibility.

To investigate mucosal immune responses to SARS-CoV-2 and estimate rates of past infection, we developed six salivary enzyme-linked immunosorbent assays (ELISA). We aimed to evaluate assay performance for detecting recently confirmed SARS-CoV-2 infection in a blinded test accuracy study. To understand salivary antibody responses further and facilitate their deployment to cohorts with unknown infection status, we correlated antibody levels measured by the assays in paired serum and saliva samples. Finally, we sought to field-test the best performing assays in a household transmission setting to investigate mucosal antibody responses in recently exposed adults and children. Our study provides robust standardisation and evaluation of saliva as a sample for SARS-CoV-2 antibody detection and provides insights into characterising mucosal immune responses following infection.

## Methods
**Study participants**. Individuals donating samples following confirmed, suspected or no SARS-CoV-2 infection were convenience samples, donated to the Bristol BioBank. 'Known positives' were those with PCR-confirmed SARS-CoV-2 infection, sampled at least 10 days post-test confirmation who responded to local, workplace advertisement. Details of symptoms, tests and other demographic and clinical information relating to the donor and their COVID-19 status were collected at the point of sampling using a case report form. Samples collected prior to SARS-CoV-2 emergence ('known negatives') were also accessed through the Bristol BioBank, alongside associated clinical and demographic information. Household members undergoing SARS-CoV-2 outbreaks were eligible to participate in the household study if one household member or close contact self-identified as SARS-CoV-2 positive (confirmed by PCR or lateral flow test). The individual that self-identified as SARS-CoV-2 positive is subsequently termed the index case. Participants were sampled as part of the CoMMinS study (COVID-19 Mapping and Mitigation in Schools; https://commins.org.uk). All household members were invited to take part for one week. If one or more saliva samples from the family were PCR positive in week 1, all participating family members were invited to continue sampling for 4 weeks. Details of symptoms, previous SARS-CoV-2 infection, vaccination history and other donor information were collected at consent using an online questionnaire, and symptoms continued to be reported throughout the sampling period alongside saliva sampling.

### Sample collection and processing
*Ethics*. Whole saliva from healthy donors (pre- and during the COVID-19 pandemic) was obtained via the Bristol BioBank (NHS REC 20/WA/0273) under the use application U-0042. Pre-pandemic (PP) sample cohorts were obtained in two ways. PP cohort 1 samples were collected in Portugal under local ethics for a specific research study, remaining samples were stored and used for this work under NHS REC 13/NW/0439. PP cohorts 2–5 were collected under further Bristol BioBank deposit applications, and upon study completion these sample sets were deposited into the Bristol BioBank and released to this project under use application U-0042. Saliva samples were collected from household outbreaks during the CoMMinS study under NHS REC 20/HRA/4876. All donors/participants provided written informed consent for the use of their samples or parental consent (with or without verbal consent), as appropriate. All samples were used in accordance with the Human Tissue Act (2004).

*COVID-19 samples*. Whole saliva was collected from individuals who had recovered from COVID-19 (PCR-confirmed infections), suspected COVID-19 cases and healthy donors through the Bristol BioBank (NHS REC 20/WA/0273). Participants were instructed to not eat/drink/brush teeth/chew gum/use mouthwash for 30 min prior to saliva collection. Participants collected their own saliva by drooling into a funnel (Isohelix, Cell Projects UK,) over the top of a sterile collection tube up to a 2 mL mark. Instructions provided included explanation of the difference between saliva and sputum. Collected specimens were promptly held at 4 °C for ≤4 h and transported to the laboratory for long-term storage at −70 °C. Peripheral blood was collected into a SST vacutainer (BD Biosciences, USA) for serum extraction. Household members likewise collected their own saliva in the CoMMinS study; the technique was described to participants by telephone and an instruction leaflet was also provided.

*Pre-pandemic samples*. Whole saliva was collected from individual's pre-pandemic and obtained via the Bristol Biobank. Full details of each pre-pandemic cohort (PP 1–5) were as follows:

*PP Cohort 1*. In March 2014, paired nasopharyngeal swab and saliva samples were collected from children (aged 4 months to 6 years) attending day care centres in Coimbra, Portugal. Saliva samples were collected using foam polygon swabs (Rocialle UK),

decanted into storage tubes and stored at −70 °C. Prior to sample collection participants were requested not to eat, drink or chew gum.

*PP Cohort 2.* During 2012–2013, fifty children aged 2–11 years were recruited to a longitudinal study, where, as part of the study, saliva samples were collected mainly using foam polygon swabs (Rocialle, UK), or some older children spat directly into a Falcon tube (Corning, USA). Saliva samples were transported at 4 °C and frozen at −70 °C within 4 h. Saliva samples were collected at baseline when the child was admitted for routine adenoidectomy or adenotonsillectomy at Bristol Royal Hospital for Children, and then monthly at five subsequent time points at their home by a Research Nurse.

*PP Cohort 3.* Between 2006–2007, healthy adults were recruited to a study in which saliva samples were collected at four time points using foam polygon swabs (Rocialle, UK).

*PP Cohort 4.* Between 2007 and 2008, thirty-two healthy adults aged 18–40 years were recruited to a study collecting saliva using foam polygon swabs (Rocialle, UK) pre and post meningococcal ACWY conjugate vaccination.

*PP Cohort 5.* In August 2019, saliva samples were collected from six healthy adults. Participants drooled into a funnel (Isohelix, Cell Projects UK) that was placed inside a collection tube. Samples were frozen within 4 h of collection at −70 °C.

### Salivary ELISA development
*Comparison of plate type.* Reactivity and background binding was compared for five different plates using the spike IgA assay: MaxiSorp, (Fisher Scientific, USA), Immulon 1B (Thermo Scientific, USA), MICROLON® plates (Greiner Bio-One, Austria), Polysorb (Thermo Scientific, USA) and Universal binding (Thermo Scientific, USA). Saliva was assayed at a single dilution (1 in 10) in duplicate on either an uncoated plate (no antigen; coated with PBS only) or coated with spike protein at 10 mg/mL. One negative (healthy donor) and one positive (clinically suspected COVID-19 donor) saliva sample were each assayed in duplicate. The plate which exhibited the lowest background binding when uncoated, as well as enhanced discrimination when coated with antigen, was selected as optimum (MICROLON® plate, Greiner Bio-One, Austria).

### Optimisation of assay conditions
Antigen coating concentration was optimised based on responses to N-protein, spike and RBD IgA by testing saliva collected from negative (healthy donor) and positive (clinically suspected or for N-protein, PCR-confirmed) donors, together with a positive serum pool (3 PCR-confirmed donors) over 4 different antigen coating concentrations: 1, 5, 10 and 20 µg/mL. Optimal antigen concentration was based on the point on the dose-response curve where the quantity of antigen saturated the plate. Checkerboard titrations were used to determine the optimum secondary antibody and sample concentration. The secondary antibody was titrated from 1 in 5,000 to 1 in 30,000; sample was diluted 3-fold from 1 in 3 to 1 in 2,430. A TMB development time of 20 min was optimised to allow for optimal discrimination between positive and negative samples and high throughput plate processing.

### Effect of heat inactivation and multiple freeze-thaw cycles on reactivity
To test the effect of inactivation on antibody signal, and the sensitivity of samples to modifications in the duration of heat inactivation, we assayed saliva samples either untreated; heat inactivated according to standard biosafety conditions: 56 °C for 30 min; or for increased durations of 56 °C for 45 min and 56 °C for 60 min. All samples were covered with parafilm during heat inactivation, centrifuged briefly to release condensation from the lid, then transferred to wet ice before returning to the freezer.

To test the effect of freeze-thawing saliva samples on antibody signal, saliva was subjected to either 2, 4 or 8 rounds of freeze-thaw. Samples of equal volume (65 µl each) were frozen at −70 °C and thawed on wet ice (~60 min) and remained thawed on wet ice for 1 h before re-freezing at −70 °C (total time thawing/thawed = 2 h).

### Conduct of immunoassays
*Sample processing.* Prior to running immunoassays, saliva was thawed on ice and centrifuged at room temperature for 5 min at 13,000 g. The supernatant was aspirated and aliquoted for heat inactivation. All saliva samples (pre-pandemic and convalescent) were heat inactivated at 56 °C for 30 min in a digital heat block (Sci-Quip, UK and Labnet USA) using validated methods.

**Saliva ELISA**. ELISAs were performed as previously described in Goenka et al[17]. Salivary antibodies specific for whole SARS-CoV-2 spike protein, for its RBD region and for the viral N-protein were detected with an ELISA based on methodology originally described for serum[1]. Modifications were made following optimisation of assay parameters described below. Final assay conditions were as follows: antigens were diluted in PBS and MICROLON® plates (Greiner Bio-One) were coated with 10 mg/mL spike protein overnight at 4 °C. Saliva supernatants were assayed singly, diluted at either 1 in 10 (IgA) or 1 in 5 (IgG) to a final volume of 100 mL per well. Secondary antibodies were used as follows with the dilution factor indicated: HRP conjugated anti-human IgG (Southern Biotech: 1 in 15,000) and IgA (Sigma: 1 in 20,000). Plates were developed with 1-StepUltra TMB-ELISA Substrate Solution (Thermo Fisher) for 20 min and the reaction was quenched with 2 M $H_2SO_4$ (Merck). All incubations were temperature controlled at 24 °C. Optical density (OD) was read at 450 nm (to measure signal) and 570 nm (background) using a BMG FLUOstar OMEGA plate reader with MARS Data Analysis software. The OD readings at 450 nm for each well were subtracted from the OD at 570 nm then corrected for the average signal of blank wells from the same plate; ODs reported are an average of duplicate wells per sample.

**Serum ELISA**. ELISAs were performed as previously described in Goenka et al[13] and Halliday et al[18], based on the methodology described previously[1]. Spike, RBD and N-protein were each diluted in sterile PBS (Sigma) and MaxiSorp plates (NUNC) were coated with either 10 mg/mL (spike) or 20 mg/mL (RBD; N-protein) of protein overnight at 4 °C before use. Plates were blocked with a 1 h incubation in 3% Bovine Serum Albumin (BSA) (Sigma-Aldrich) in PBS with 0.1% Tween-20 (Sigma-Aldrich) (PBS-T) at room temperature. Serum samples were thawed on ice before use, tested in duplicate and diluted to a final volume of 100 µL per well at a pre-optimised dilution, either at 1 in 50 (IgA) or 1 in 450 dilution (IgG), in dilution buffer (1% BSA in PBS-T). All samples were tested on a single plate for each antigen and antibody isotype combination. Secondary antibodies were used as follows with the dilution factor indicated: HRP conjugated anti-human IgG (Southern Biotech: 1 in 25,000) and IgA (Sigma: 1 in 6,000- 10,000). SIGMA FAST ™ OPD (o-phenylenediamine dihydrochloride) (Sigma-Aldrich) was used to develop plates and reactions were stopped after 30 min with 3 M HCl. ODs were read at 492 nm and 620 nm using the same reader used for salivary ELISAs.

**Production of protein for ELISA**. Production of antigens was performed according to methods performed and described previously[17]. SARS-CoV-2 trimeric spike protein ectodomain and the RBD of the spike protein were produced in insect cells as described[19]. The spike construct consists of amino acids 1 to 1213 and with a C-terminal thrombin cleavage site, a T4-foldon trimerization domain followed by a hexahistidine tag for affinity purification. The polybasic cleavage site has been removed (RRAR to A) in this construct[19]. RBD from spike protein was also produced as described in Toelzer et al.[19]. This construct contains SARS-CoV-2 spike amino acids R319 to F541, preceded by the native spike signal sequence (amino acid sequence MFVFLVLLPLVSSQ) at its N-terminus and followed by a C-terminus octa-histidine tag for purification. A codon-optimised, N-terminal His6 tagged full length nucleocapsid protein of SARS-CoV-2 was synthesised and cloned by GenScript into a pET28a bacterial expression plasmid, (called here pET28a-NP-FL). The pET28a-NP-FL plasmid was transformed into *E. coli* strain BL21 (DE3) and expressed.

**Pooled sera and saliva QC material**. To facilitate assay standardisation and longitudinal monitoring of results, a serum standard pool of known antibody level was run on all serum and saliva ELISA plates. 'High' and 'low' saliva quality control pools were run on all saliva ELISA plates to monitor assay variation. The saliva high control pool was generated using large sample volumes collected from three individuals with PCR-confirmed COVID-19 infections. The low saliva control pool was generated from saliva from two healthy donors who had no known COVID-19 infection history and low antibody levels on all assays. Inter-assay variation was also monitored in serum ELISA using two serum standards of differing antibody levels. The serum standard was generated by combining sera from 3 individuals with PCR-confirmed COVID-19 infections. Aliquots of standards were created and stored at −70 °C to ensure consistent performance.

**Threshold setting and evaluation of assay performance in a prospective test accuracy study**. The test accuracy component of this study is reported following STARD guidelines. The completed STARD checklist is given in Supplementary Table S1. See Supplementary Tables S2 and S3 for STARD Appendix.

**Allocation of samples to the threshold and validation set**. Sample numbers were decided by the availability of samples required to address the study aims, with awareness of MHRA guidance stipulating a requirement of at least 200 confirmed positive cases and 200 confirmed negative cases to estimate ≥98% sensitivity and ≥98% specificity[20]. Saliva samples collected pre-pandemic (known negatives) and from recent PCR-confirmed cases (known positives) were spilt 50:50 across two sample sets: a threshold set, used to determine thresholds for positivity and a validation set, for evaluating assay performance. A total of 346 saliva samples belonging to 228 unique donors, of which 52 donors had repeat samples were considered in allocations. We assigned 84/346 (24.3%) of these samples to the threshold set as they were assayed during assay development. Samples not assayed as part of development were randomised to the threshold set so that 50% of total cases and 50% of total controls appeared in threshold and validation sets. Stratified random sampling resulted in the following threshold set allocation: asymptomatic PCR-confirmed (N = 4); symptomatic PCR-confirmed (N = 12); adult pre-pandemic (N = 22) and child pre-pandemic (N = 61). The final allocation of samples and characteristics in the threshold and validation set is shown in Table 1.

**Setting thresholds for positivity**. Threshold set samples (N = 160) were assayed in a four-point 3-fold dilution series singly starting at either 1 in 10 for IgA or 1 in 5 for IgG against N-protein, RBD and spike. Discrimination between positive and negative samples by each antigen/secondary combination was largely independent of dilution and discrimination was slightly improved at higher concentrations without reaching saturation, thus informing proceeding with the top dilution in validation set testing. Receiver operator characteristic (ROC) curves were constructed for each of the 6 assays using threshold set samples and four thresholds were set: those to achieve 97%, 98% and 99% specificity among the known negative population, and that which maximised the Youden's index. ROC curves were used to evaluate trade-offs in sensitivity and specificity of threshold set samples. After evaluating each assay's overall performance in the combined threshold/validation set, the threshold which provided optimal detection of PCR-confirmed cases (i.e., highest sensitivity) whilst maintaining at least 98% specificity in validation was selected.

**Estimation of test accuracy**. Validation set samples (N = 160) were assayed in a blinded fashion at a single point dilution in duplicate (1 in 10 for IgA; 1 in 5 for IgG). Clinical information and index test results were not available to the assessors of the reference standard. RBD IgA and IgG assays were dropped from evaluation due to poor performance in the threshold set. Performance was evaluated for the N-protein and spike IgA and IgG assays using ROC curve analysis on validation set samples only, or to increase precision, threshold and validation set samples combined (N = 320). A sensitivity analysis was performed comparing the validation vs full sample set to assess the impact of combining samples on performance estimates. Individuals with multiple samples were not de-duplicated and all samples were included in estimates of test accuracy. A sensitivity analysis was performed comparing estimates of assay performance (AUC, specificity and sensitivity) including all samples (i.e., the primary analysis) with results based on analysis of the first sample donated by each individual only. Repeat samples from the same donor were found to have little impact on test performance in sensitivity analysis, so all samples were included to estimate test accuracy accordingly (Tables S5 and S6). There were no indeterminate index or reference standard test results since this was not a category, test results were either positive or negative. Samples with volumes too low to assay were excluded from ROC analysis. Positive and negative predictive values at population prevalence's of 0.1, 1, 5, 10, 20 and 40% previous SARS-CoV-2 infection were modelled. The variability in diagnostic accuracy was assessed by examining the association of false positivity with age and sex, and false negativity by time since symptom onset and symptom status (categorised as asymptomatic; 11–21 days post symptom onset; 22–43 days, 44–70 days; and ≥71 days).

**Correlating mucosal and systemic antibody**. To investigate salivary and serum responses in paired samples, serum samples for which saliva was collected on the same day were assayed for antibody specific for SARS-CoV-2. Due to low sample volumes, the final number of samples tested for each of the 6 assays differed: spike protein IgA = 97 and IgG = 81; RBD IgA = 35 and IgG = 33; N-protein IgA = 91 and IgG = 80.

**Detection of SARS-CoV-2 infection by RT-qPCR on saliva**. Saliva samples collected in household outbreaks were tested for the presence or absence of SARS-CoV-2 using a PCR protocol that was developed and optimised in-house. In brief, a 90 μl aliquot of each neat saliva sample was chemically lysed using L6

**Table 1 Demographic characteristics of samples used in assay development and evaluation.**

| | PCR-confirmed case[a] | | Pre-pandemic | | Suspected case[b] | Healthy donor[c] |
|---|---|---|---|---|---|---|
| | Threshold (N = 45) | Validation (N = 45) | Threshold (N = 115) | Validation (N = 115) | Optimisation (N = 10) | Optimisation (N = 16) |
| *Sex* | | | | | | |
| Female | 33 (73.3%) | 34 (75.6%) | 48 (41.7%) | 39 (33.9%) | 6 (60.0%) | 11 (68.8%) |
| Male | 12 (26.7%) | 11 (24.4%) | 67 (58.3%) | 76 (66.1%) | 4 (40.0%) | 5 (31.3%) |
| *Age (years)* | | | | | | |
| Mean (SD) | 37.7 (12.0) | 42.5 (15.5) | 10.8 (10.6) | 10.6 (9.92) | 39.0 (9.83) | 37.3 (14.2) |
| Median [Min, Max] | 37.0 [18.0, 64.0] | 46.0 [18.0, 68.0] | 5.00 [2.00, 39.0] | 5.00 [1.00, 39.0] | 37.0 [27.0, 53.0] | 31.5 [23.0, 65.0] |
| Unknown | 0 (0%) | 2 (4.4%) | 3 (2.6%) | 0 (0%) | 0 (0%) | 0 (0%) |
| *Adult/Child* | | | | | | |
| Adult | 45 (100%) | 45 (100%) | 36 (31.3%) | 35 (30.4%) | 10 (100%) | 16 (100%) |
| Child | 0 (0%) | 0 (0%) | 79 (68.7%) | 80 (69.6%) | 0 (0%) | 0 (0%) |
| *Days post-symptom onset* | | | | | | |
| Mean (SD) | 66.1 (35.5) | 52.0 (18.9) | NA | NA | 93.5 (48.7) | 146 (67.9) |
| Median [Min, Max] | 59.0 [11.0, 133] | 47.0 [15.0, 107] | NA | NA | 84.0 [42.0, 208] | 163 [26.0, 216] |
| Unknown | 5 (11.1%) | 4 (8.9%) | 115 (100%) | 115 | 0 (0%) | 10 (62.5%) |
| *COVID-19 symptoms* | | | | | | |
| Asymptomatic | 5 (11.1%) | 4 (8.9%) | 0 (0%) | 0 (0%) | 0 (0%) | 10 (62.5%) |
| Symptomatic | 40 (88.9%) | 41 (91.1%) | 0 (0%) | 0 (0%) | 10 (100%) | 6 (37.5%) |
| Unknown | 0 (0%) | 0 (0%) | 115 (100%) | 115 (100%) | 0 (0%) | 0 (0%) |

A total of 320 samples collected pre-pandemic (N = 230) and from PCR-confirmed SARS-CoV-2 cases (N = 90) were randomised across threshold and validation sample sets. Additional samples used in assay optimisation (N = 26) were collected from suspected cases and healthy donors.
[a]PCR-confirmed cases were diagnosed between March to November 2020.
[b]Suspect case: symptomatic with epidemiological link, laboratory unconfirmed;
[c]Healthy donor: no SARS-CoV-2 history or symptoms.

Lysis Buffer (20-8600-15, Severn Biotech Ltd.). A MS2 RNA bacteriophage internal control was added, and samples were extracted using the QIAsymphony SP automated system (QIA-GEN) or KingFisher Flex Purification System (ThermoFisher Scientific) following the manufacturers' instructions. Total nucleic acid was eluted in 60 μl or 50 μl of which 10 μl was used in RT-qPCR using the SARS-CoV-2 N6/E and MS2 probe and gene primers (Metabion). SARS-CoV-2 E gene primers and probe were as previously described[21]. SARS-CoV-2 N6 gene primers and probes were designed using Primer3 and a consensus multiple sequence alignment of 658 SARS-CoV-2 N gene sequences downloaded from GenBank2[22]. Each PCR reaction well contained 6.25 μl of TaqPath 1- Step RT-qPCR Master Mix, CG (ThermoFisher Scientific), 1 μl of 25X primer and probe mix, 7.75 μl of molecular grade water and 10 μl of total nucleic acid extract. The QuantStudio 7 Real-Time PCR System (Applied Biosystems) was used for RT-qPCR where thermal cycling consisted of: 25 °C for 2 mins, 50 °C for 15 min and 40 cycles of 95 °C for 10 s, 60 °C for 30 seconds. Samples producing a cycle threshold (Ct) ≤35 were considered positive. Full sequences and final concentrations of primers and probes used are given in Table S4.

**Case definition.** For assay development and test accuracy, healthy donors self-reported no SARS-CoV-2 history or symptoms; suspected cases reported symptoms with an epidemiological link but SARS-CoV-2 infection unconfirmed; PCR-confirmed cases reported a RT-qPCR positive test performed on a nose/throat swab through NHS testing; pre-pandemic controls were collected at least 6 months prior to SARS-CoV-2 emergence. For analysis of household outbreaks, we categorised index cases and household contacts into PCR positive or PCR negative at any point in the study. PCR was performed on the same saliva sample tested for antibody; positivity was set on a Ct value ≤35. Index cases were those that originally self-reported a positive PCR or LFT result, and on enrolment had two consecutive PCR positives.

**Statistics and Reproducibility.** All statistical analyses were performed using the R-studio environment, with the library 'tidy-verse' for data manipulation and summary statistics, 'pROC' for ROC analysis and 'binom' for estimating binomial confidence intervals. The libraries 'ggplot2', 'patchwork', 'cowplot' and 'ggstatsplot' were used for data visualisation. Antibody levels were expressed as a normalised optical density (Norm OD) by dividing the mean background-corrected OD of duplicate test samples by the mean background-corrected OD of the duplicate top dilution of the standard. Assay reproducibility was assessed by calculating the coefficient of variation for controls tested in duplicate on the same plate (intra-assay variation) and between plates (inter-assay variation) using plates run in the household study. 95% confidence intervals for AUC (area under the receiver operating characteristic curve) were calculated using DeLong's method[23] or computed with 10,000 stratified bootstrap replicates for sensitivity and specificity estimates. Antibody responses were compared across multiple groups using the Kruskal-Wallis test with post-hoc testing using Dunn's test. A Bonferroni correction was applied for multiple pairwise comparisons. Significance was defined as $p \leq 0.05$. Kendall's Tau correlation coefficient and associated $P$ value were calculated for salivary and serum antibody correlations.

AdaBoost classifiers were trained to predict positive and negative individuals and model performance was measured by calculating ROC AUC scores. Model training and testing were performed as part of a 5-fold cross-validation loop. AdaBoost classifiers were trained on 13 datasets in total—6 models were trained with individual assays; 7 models were trained with a combination of assays. Datasets were constructed using samples in the combined threshold and validation sample set (1 in 5 for N-protein, RBD and spike IgG; 1 in 10 for N-protein, RBD and spike IgA; N = 318). To ensure enough data for cross-validation, we limited assay combinations tested to those containing either the same antigen or secondary antibody. We also tested combining only the N-protein and spike antigens, as RBD assays were not taken forward for threshold setting nor to the field

studies. This meant that the sizes of the datasets we used for model training and testing were, depending upon the assay(s) included, in the range of 92–150 individuals. The AdaBoost algorithm was imported into the notebook from the Python package scikit-learn[24].

To determine rates of salivo-positivity in the household study, the proportion of individuals with antibody above the threshold for positivity (final thresholds given in Table 2) were divided by the total number of individuals sampled, stratified by infection status (PCR positive/negative during the study) and/or vaccination. Rates of salivo-conversion were calculated based on an individual becoming antibody positive following antibody negativity at Day 0; those who were antibody positive at Day 0 were removed from the denominator.

**Reporting summary**. Further information on research design is available in the Nature Portfolio Reporting Summary linked to this article.

## Results

**Development of salivary immunoassays for SARS-CoV-2 antibody detection**. Single-dilution salivary ELISAs capable of detecting antibodies specific to SARS-CoV-2 full length spike protein, RBD and N-protein were developed based on previously described methodology for serum[1] (Fig. 1). Assay operating conditions were optimised to reduce background, achieve maximum discrimination between positive and negative samples and retain a good dynamic range (Fig. S1). We observed highest background on a high-binding hydrophilic plate (NUNC Maxisorp): on the remaining medium binding, hydrophobic plates, background was low and comparable, although optimum discrimination was found using the Greiner plate (Fig. S1a). Optimal antigen coating concentration was determined to be 10 µg/ml for each antigen (Fig. S1b). Using checkerboards, we determined optimum secondary antibody and sample concentrations (IgA: 1 in 10 saliva starting dilution and 1:20,000 secondary IgA antibody; IgG: 1 in 5 saliva starting dilution and 1:15,000 secondary IgG antibody, Fig. S1c, d). Heat inactivation (56 °C for 30, 45 or 60 min) and freeze-thawing of samples (2, 4 or 8 cycles) did not affect ELISA signal (Fig. S2), allowing for safe and practical handling of samples.

The demographics and clinical characteristics of 230 known negative (pre-pandemic) and 90 known positive (convalescent SARS-CoV-2 PCR-confirmed) donors are given in Table 1. Samples were randomised 50:50 across two sample sets: a threshold set, used to determine thresholds for positivity and a validation set, for evaluating assay performance, each containing samples from 115 known negative and 45 known positive individuals. Due to low sample volumes, the final number of samples tested for each of the 6 assays differed (Table S7). The distribution of antibody responses and associated performance for the 6 assays obtained on threshold set samples is shown in Figs. 2a–i and S3. Based on the threshold set, the spike assays performed best out of all antigens, as shown by the highest area under the curve (AUC): IgA (92.2%, CI 95%: 88.1–96.3) and IgG (94.9%, CI 95%: 91.6–98.3) (Table S7). Discrimination between pre-pandemic and PCR-confirmed samples was poorer for N-protein and RBD assays compared to spike (Fig. S3), reflected in lower performance estimates (AUC: 60.0–85.9%, Table S7). All assays showed high levels of reproducibility as assessed by low intra- and inter-assay signal variation in internal serum and saliva controls (Tables S8 and S9).

**Evaluation of diagnostic performance for salivo-surveillance**. Next, we evaluated assay performance in the blind validation set

**Table 2 Evaluation of assay performance.**

| Assay | Pre-pandemic (N) | PCR-confirmed (N) | Threshold method | Threshold value | AUC (%) | 95% CI | False positives (N) | Specificity (%) | 95% CI | False negatives (N) | Sensitivity (%) | 95% CI |
|---|---|---|---|---|---|---|---|---|---|---|---|---|
| N-protein IgA | 173 | 81 | 97th percentile | 0.524 | 71.9 | 65.7–78.1 | 6 | 96.5 | 93.6–98.8 | 74 | 8.64 | 3.7–14.8 |
| N-protein IgG | 173 | 80 | 98th percentile | 0.484 | 84.6 | 79.9–89.4 | 3 | 98.3 | 95.9–100 | 66 | 17.5 | 10–26.2 |
| Spike IgA | 223 | 87 | 99th percentile | 0.384 | 89.9 | 86.5–93.2 | 5 | 97.7 | 95.5–99.5 | 55 | 36.8 | 26.4–47.1 |
| Spike IgG | 197 | 83 | 98th percentile | 0.306 | 95.0 | 92.8–97.3 | 3 | 99.0 | 97.4–100 | 41 | 50.6 | 39.8–61.4 |

Performance of N-protein and spike IgA and IgG assays was evaluated using known negative (pre-pandemic) and known positive (PCR-confirmed) samples in combined threshold and validation sets combined. Normalised OD was used as input value in ROC analysis. AUC = Area under the ROC curve. 95% CI = 95% confidence interval calculated for AUC using DeLong's method or computed with 10,000 stratified bootstrap replicates for sensitivity and specificity estimates.

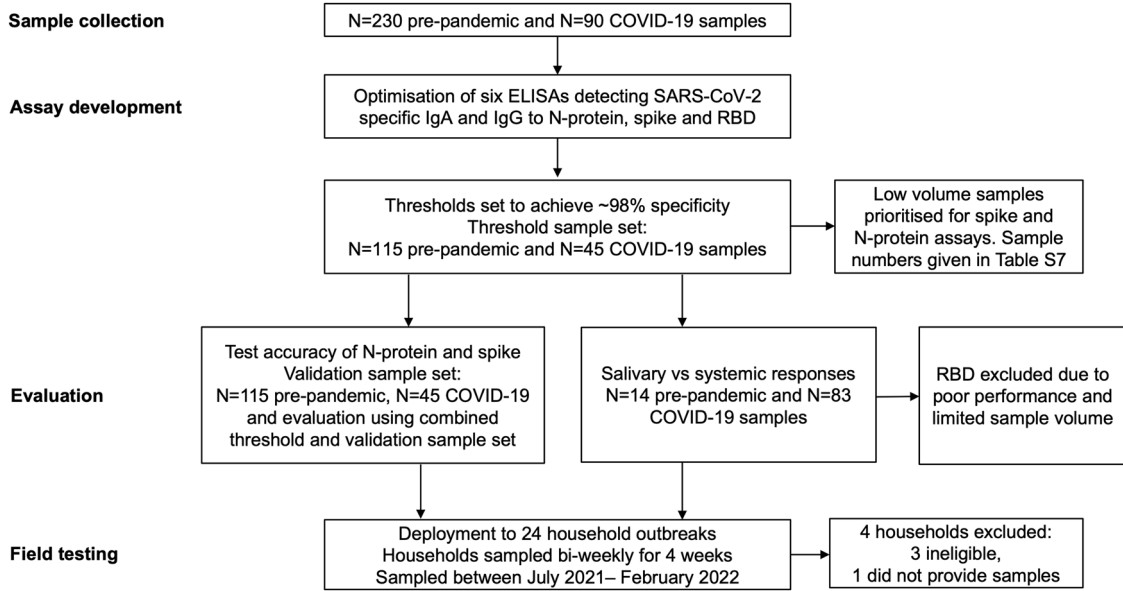

**Fig. 1 Study flow diagram describing samples and the processes used in immunoassay development, evaluation and field-testing.** Saliva and serum samples were collected pre-SARS-CoV-2 emergence from children and adults (known negatives, $N = 230$), and from individuals with PCR-confirmed SARS-CoV-2 infection (known positives, $N = 90$). Known negative and positive samples were used to set thresholds for positivity and evaluate performance. Assays were field tested using samples ($N = 510$) from 20 household outbreaks.

using the four pre-defined thresholds determined from the threshold set (Table S10 and Fig. S3). Since estimates of test accuracy (AUC, sensitivity and specificity) were similar in the validation or combined threshold/validation sample set (Table S10), we present overall accuracy estimates based on the full combined sample set, to increase precision (Table 2 and Fig. 2). The clearest discrimination between known negative and positive samples was shown for the spike IgA and IgG assays (Fig. 2c, f, i). Poor discrimination was observed for both N-protein (Fig. 2a) and RBD IgA (Fig. 2b): known positives showed reactivity to N-protein and known negatives exhibited reactivity to RBD. Given the poor performance of the RBD assays (low specificity and sensitivity) in the threshold setting phase, and limited sample volume, RBD was dropped from evaluation. The best-performing assays were spike IgG (across both sample sets combined: AUC 95.0%, 95% CI: 92.8–97.3%) and spike IgA (AUC 89.9%, 95% CI: 86.5–93.2%), followed by N-protein IgG (AUC 84.6%, 95% CI: 79.9–89.4%). N-protein IgA had the poorest performance (AUC 71.9%, 95% CI: 65.7–78.1%, Table 2). We observed that thresholds set for 98% specificity in the threshold set maintained this performance in the validation set (Table S10, Figs. S4 and S5). The highest sensitivity observed was for spike IgG (50.6%, 95% CI: 39.8–61.4%) and lowest sensitivity for N-protein IgA (8.6%, 95% CI: 3.7–14.8%) (Table 2). Taken together, primary infection with SARS-CoV-2 induces salivary antibody responses against spike IgA and IgG, whereas the N-protein and RBD responses were restricted largely to IgG.

Considering false positivity by age, specificity was 100% in younger age groups (0–19 years) for spike IgA, IgG and N-protein IgG (10–19 years) assays. Specificity was lower for N-protein IgA (0–9 years: 98.2%; 10–19 years: 88.9%) and N-protein IgG (0–9 years: 98.2%). The lowest observed specificity was for spike IgG in 30–39 years (81.8%) (Table S11). There was no indication of specificity varying by sex for N-protein or spike (Table S12), which has been reported previously[25,26]. In general, sensitivity declined with increasing time since symptom onset for all assays (Table S13). No PCR-confirmed asymptomatic cases were N-protein positive (0/8): sensitivity was higher in these asymptomatic cases for spike IgA (33.3%) and IgG (11%). For all assays, no clustering in antibody responses were observed for pre-pandemic samples tested from various collections: however, signal was statistically increased for adults compared to children (Fig. S6). Positive predictive value (PPV) at 5% prevalence was higher (fewer false positives) in the spike IgA (30.6–45.7%), IgG (66–82.4%) and N-protein IgG (30.1–35.1%) assays compared to N-protein IgA (9.8–11.5%). NPV was lowest (increased false negatives) for N-protein IgA (95.1–95.3%). Ranges for PPV and NPV indicate values at each threshold method (97th to 99th percentile and Youden's index), estimates were robust up to 40% prevalence (Fig. S7).

**Combining assays to predict positive and negative individuals.** There was both heterogeneity and discordance in the isotype response for the same antigen: responses were predominantly SARS-CoV-2 specific IgA or IgG, few individuals had high levels of both isotype (Fig. 3a). Consistent with our earlier analysis (Fig. 2c, f), the spike IgG, and to a lesser extent the spike IgA assay achieves the best discrimination. Given the heterogeneity in response, we speculated that combining readings across multiple assays could improve sensitivity for recent infection. To test this hypothesis, we trained AdaBoost classifiers[27] to predict positive and negative individuals using either one or a combination of the 6 assays. The best-performing model was trained with data from the N-protein, RBD and spike IgG assays (mean ROC AUC score = 0.94; Fig. 3b). The performance of this model was substantially better than the performances of the models trained with the individual assays (mean ROC score between 0.54 for N-protein IgA and 0.82 for spike IgG). The model trained with the spike IgA and IgG assays (mean ROC AUC score = 0.88) performed somewhat better than those trained using the individual spike IgA (mean ROC AUC score = 0.76) and spike IgG (mean ROC AUC score = 0.82) assays (Fig. 3b). Combining N-protein IgG and spike IgG assays (mean ROC AUC score = 0.89) gave very similar performance to combining both spike assays. The performance of all models based on assays individually or combined is shown in Table S14.

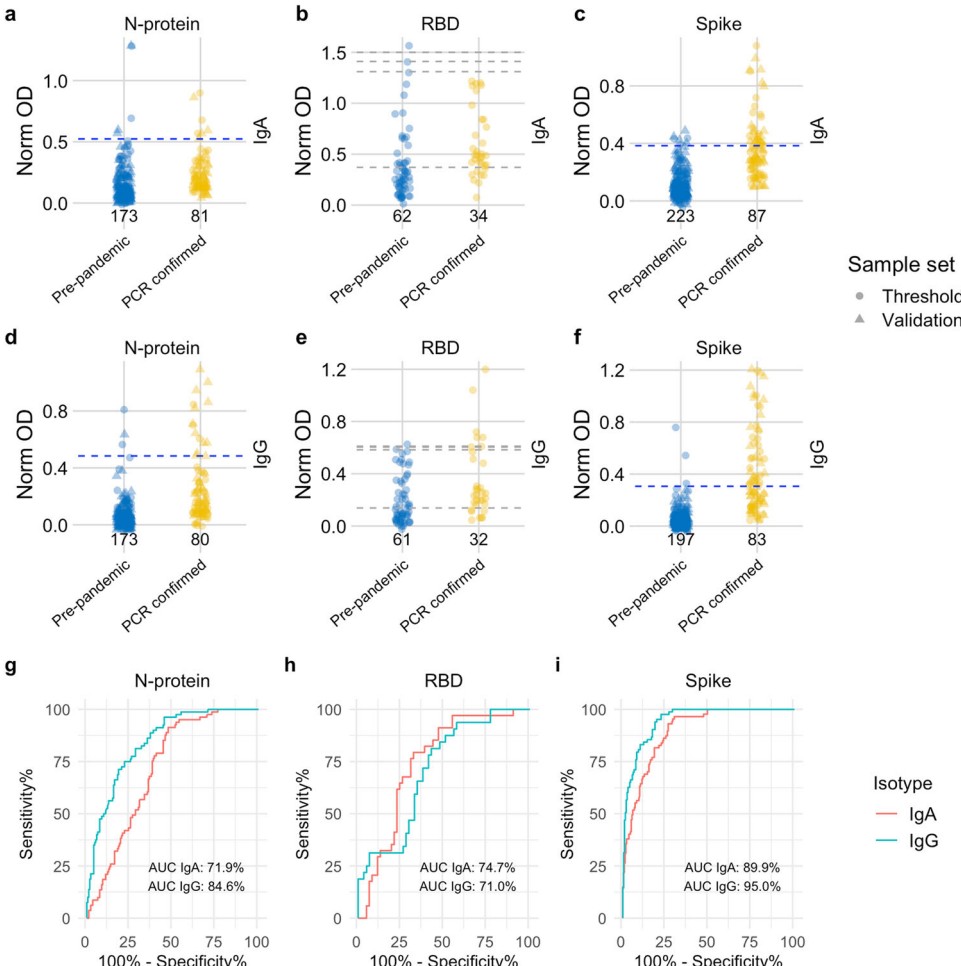

**Fig. 2 Distribution of semi-quantitative titres for each assay and corresponding ROC curves. a–c** Dotplots show the scatter of values for SARS-CoV-2 specific IgA responses against N-protein ($N = 254$), RBD ($N = 96$) and spike ($N = 310$), specific IgG responses against N-protein ($N = 253$), RBD ($N = 93$) and spike ($N = 280$) are shown in (**d–f**). Data are presented for known negative and known positive samples, the number of samples tested in each group is also shown under the corresponding data points. For N-protein and spike, proposed thresholds (97th–99th percentile and Youden's index) are shown as dashed lines, with the final selected threshold in blue; thresholds were derived in the threshold setting phase (threshold samples shown as circles, validation samples shown as triangles). RBD was not taken forward to full evaluation, so performance in the threshold set and corresponding proposed thresholds are shown as dashed grey lines. **g, i** ROC curves for N-protein and spike represent assay performance determined on all threshold and validation set samples combined. **h** The ROC curve for RBD represents assay performance in threshold sample set. ROC = receiver operating characteristic curve. AUC = area under the curve. Norm OD = Normalised OD (a relative ratio to the serum standard). N-protein = Nucleocapsid protein. RBD = Receptor binding domain.

**Salivary IgA antibody indicates mucosal antibody responses and IgG, systemic antibody responses.** Salivary antibody responses were compared with serum antibody to investigate the mucosal immunological profiles in individuals with recent SARS-CoV-2 infection (Fig. 4). Among the 320 available samples, 97 individuals had had saliva and serum collected on the same day, of whom 83 were PCR-confirmed and 14 pre-pandemic (see Methods for further details). Results from samples collected from PCR-confirmed cases were positively correlated for all 6 assays, but all the IgA assays were less well correlated between saliva and serum than the IgG assays (Tau = 0.11, 0.23, 0.22: Tau = 0.58, 0.33, 0.39 N-protein, RBD and spike IgA and IgG, respectively), with several individuals having specific salivary IgA in the absence of detectable serum IgA antibody. N-protein IgG responses exhibited the strongest positive saliva-serum correlation (Tau = 0.58, $p < 0.001$, $n = 73$), whereas N-protein IgA exhibited the weakest correlation (Tau = 0.11, $p = 0.14$, $n = 78$). Fewer matched pre-pandemic samples were available but are plotted for visual reference. For salivary samples assigned as

positive for spike IgA ($N = 28$) or spike IgG ($N = 40$) based on validated thresholds, we explored the distribution of antibody responses in relation to time since symptom onset and age, and how these salivary responses correlated to serum (Fig. S8). Salivary antibodies were detectable up to 123- and 133-days post onset of symptoms for spike IgA and IgG, respectively. Trends were similar between the two sample types for both isotypes and there were no marked apparent differences associated with age or time since symptom onset, although sample sizes were small.

**Field-testing assays in SARS-CoV-2 household outbreaks.** Spike and N-protein assays were deployed on samples collected following recent household transmission events (13/07/2021 to 22/02/2022) to evaluate their utility in monitoring SARS-CoV-2 infections under field conditions (when Delta and Omicron variants were prevalent in the UK). Twenty households consisting of 19 index cases (10 children and 9 adults) and 48 household contacts self-sampled twice weekly for 4 weeks (Fig. 1 and Table S15). Note one index case did not provide sufficient saliva

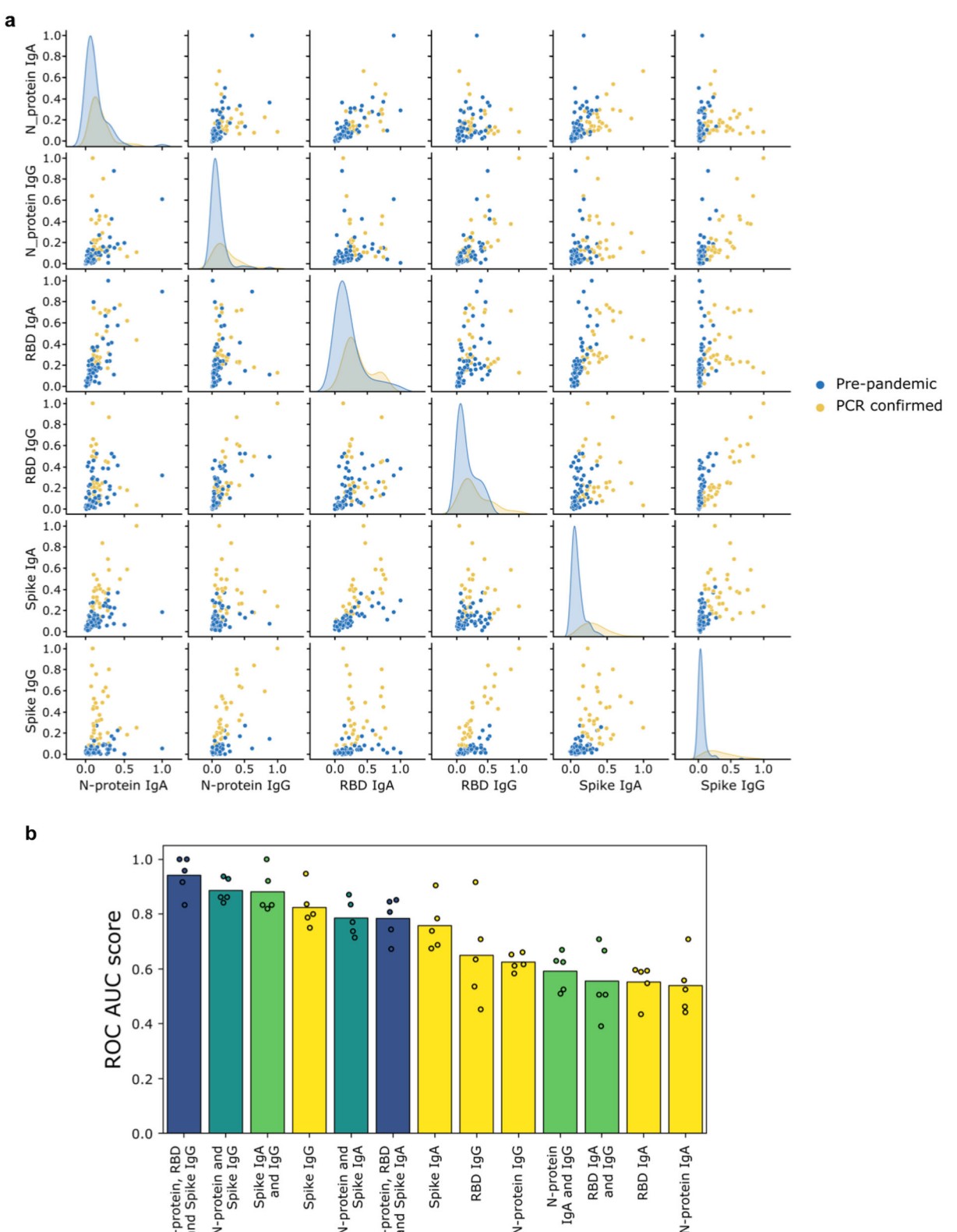

for analysis. All households included at least 1 child. All 19 index cases and 11 contacts were PCR positive on Day 0 (prevalent infection); 5 contacts became PCR +ve during the study (incident infection); and 28 contacts remained PCR -ve (see Fig. S9 for viral shedding profiles). Of the PCR +ve cases, 23/36 (63.9%) reported symptoms. Four participants reported a previous PCR-confirmed infection (between 30 to 73 days prior to Day 0). One vaccinated individual reporting prior infection was re-infected during the study when Omicron was dominant (January 2022).

Most PCR +ve cases (34/36, 94.4%) mounted salivary spike IgA or IgG responses, whilst fewer than half raised antibodies to N-protein (Table 3). Of the PCR +ve cases that were

**Fig. 3 Combining assays improves discriminatory performance. a** Pairwise scatter plots and kernel density estimates of antibody responses for N = 229 pre-pandemic (blue) and N = 89 SARS-CoV-2 PCR-confirmed (orange) samples assayed at a single dilution in each of the 6 assays: spike, N-protein, RBD IgA (1 in 10) and IgG (1 in 5). The kernel density estimates along the diagonal represent the distribution of responses measured for a single assay, whilst the scatter plots compare the responses measured across two different assays. **b** Comparison of the performance (measured via ROC AUC score) of AdaBoost models trained either with one of the 6 individual assays (yellow bars and dots), or with a selected combination of those assays (green, turquoise, blue bars and dots). The models were trained using 5-fold cross-validation: the dots represent the ROC AUC scores measured for the individual folds, whilst the bars represent the mean of these 5 scores. ROC AUC score = area under the receiver operating characteristic curve.

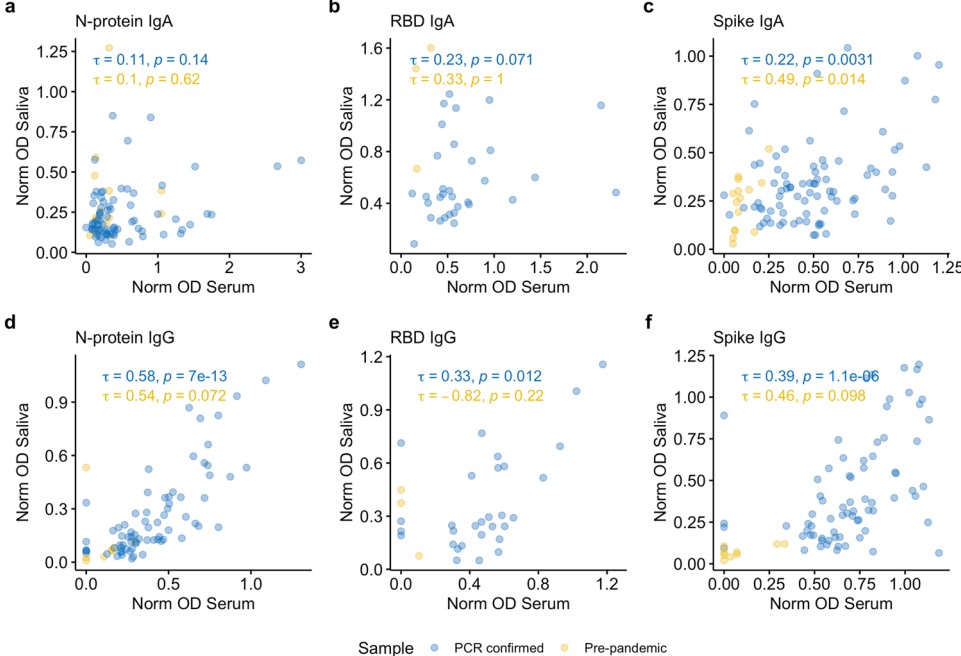

**Fig. 4 Correlation between mucosal and systemic antibody. a–f** Correlation between serum and salivary IgA and IgG responses to spike protein, nucleocapsid protein (N-protein) and receptor binding domain (RBD) **a, d** N-protein IgA (N = 91) and IgG (N = 80) assays. **b, e** RBD IgA (N = 35) and IgG (N = 33), **c, f** spike IgA (N = 97) and IgG (N = 81). PCR-confirmed samples are shown in blue and pre-pandemic samples are shown in yellow. Correlations (Kendall's tau) were performed for paired saliva and serum samples collected on the same day.

asymptomatic, most were antibody positive (11/13, 84.6%). The two PCR +ve cases who did not have detectable specific antibody were asymptomatic children (<10 years) and were PCR +ve on Day 0 only. In this setting, combining IgA and IgG results for both antigens increased sensitivity for PCR +ve cases, although no improvement was seen when combining antigens, as the few individuals that raised anti-N-protein antibody had also raised anti-spike antibody (Table 3). Spike antibody positivity detected ongoing household infections, as rates of anti-spike (IgA or IgG) increased through the 26-day period in PCR +ve cases but remained relatively constant among PCR -ve contacts (Fig. 5b, d). Similarly, for all assays, rates of salivo-conversion (i.e. antibody negative at Day 0 and positive on at least one timepoint subsequently) were higher for PCR +ve cases than PCR –ve contacts (Table 4). For example, spike IgG conversion rates were 79.2% and 27.8% for PCR +ve and PCR -ve household members respectively. Spike and N-protein antibody was detected among some PCR +ve cases and PCR -ve contacts on Day 0, suggestive either of pre-existing antibody or early mucosal responses generated post exposure/infection shortly before study enrolment (Fig. 5a–d).

In the context of different prior exposures (vaccination and/or infection), vaccinated PCR +ve individuals (mostly adults)

**Table 3 Detection of salivary antibody in household members.**

| Assay | PCR +ve cases (N = 36) n (%) | PCR –ve contacts (N = 31) n (%) |
|---|---|---|
| Spike IgA | 23 (63.9%) | 13 (41.9%) |
| Spike IgG | 31 (86.1%) | 18 (58.1%) |
| N-protein IgA | 14 (38.9%) | 11 (35.5%) |
| N-protein IgG | 6 (16.7%) | 2 (6.5%) |
| N-protein IgA or IgG | 16 (44.4%) | 13 (41.9%) |
| Spike IgA or IgG | 34 (94.4%) | 22 (71.0%) |
| Spike IgA or N-protein IgA | 25 (69.4%) | 15 (48.4%) |
| Spike IgG or N-protein IgG | 32 (88.9%) | 18 (58.1%) |
| Spike or N-protein, IgA or IgG | 34 (94.4%) | 22 (71.0%) |

The number and proportion (%) of PCR-confirmed cases (N = 36) and PCR negative contacts (N = 31) with antibody above thresholds for positivity on at least one occasion measured by N-protein and spike IgA and IgG assays.

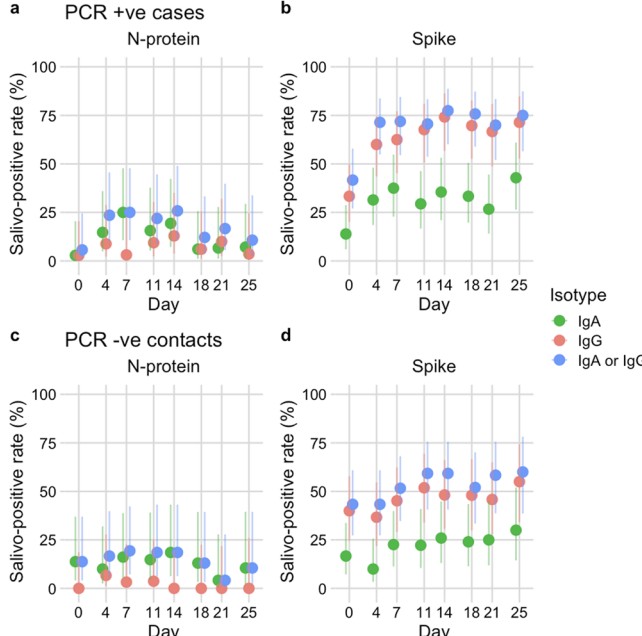

**Fig. 5 Application of assays to household outbreaks. a-d** Salivo-positive rate for 20 households with SARS-CoV-2 outbreaks (N = 20 index cases and 51 contacts) based on detection of anti-nucleocapsid (N-protein) or anti-spike, IgA (green points), IgG (red points), IgA or IgG (blue points). Salivo-conversion for household members with PCR-confirmed infection (N = 31) during the study is shown in (**a**, **b**), household members remaining PCR negative (N = 36) are shown in (**c**, **d**). Error bars represent 95% confidence intervals for a single proportion (Wilson method).

**Table 4 Rates of salivo-conversion during the household study.**

| Assay | PCR +ve cases n/N (%) | PCR –ve contacts n/N (%) |
|---|---|---|
| Spike IgA | 18/31 (58.1%) | 7/25 (28.0%) |
| Spike IgG | 19/24 (79.2%) | 5/18 (27.8%) |
| N-protein IgA | 13/34 (38.2%) | 6/25 (24.0%) |
| N-protein IgG | 5/33 (15.2%) | 1/29 (3.4%) |

The number and proportion (%) of PCR-confirmed cases and PCR negative contacts that salivo-convert during the study i.e., antibody negative at Day 0 and antibody positive thereafter.

exhibited the highest rates of antibody positivity: all cases were positive for spike IgG (20/20, 100%) and 70% were positive for spike IgA (14/20) (Fig. S10 and Table S16). The highest rates of positivity to N-protein IgA or IgG were seen in this group (12/20, 60.0%). All unvaccinated individuals were children (27/67, 40.3%), those remaining PCR -ve mostly raised spike and N-protein IgA antibody responses (4/11, 36.4%), with one individual also spike IgG positive. Unvaccinated PCR +ve children predominately raised antibody to spike IgA or IgG (14/16, 87.5%), with fewer N-protein IgA or IgG positive (4/16, 25.0%).

## Discussion

In this study, we demonstrate that saliva (spit) samples can easily be collected and used reliably to detect recent SARS-CoV-2 infection in children and adults via the measurement of SARS-CoV-2 specific antibodies. In an unvaccinated population, we found assays measuring responses to the spike protein provided better discrimination between known negative (pre-pandemic) and known positive (PCR-confirmed) samples than anti-RBD and N-protein assays. However, sensitivity is considerably reduced compared to conventional serum-based tests. Machine learning analyses suggested that combining assays detecting the same antibody isotype against different antigens (N-protein, RBD and spike), particularly IgG, can further improve diagnostic performance, and to a lesser extent combining anti-spike IgA and IgG assays likewise. As expected, our observations suggest that detectable salivary IgA largely reflects mucosal immune responses following infection, whereas IgG may primarily reflect systemic immune responses. When field tested in household outbreaks, salivary antibody responses were a reliable indicator of recent infection and exposure. Our methods and results support the importance and feasibility of using saliva as a mucosal sample for monitoring SARS-CoV-2 infection and immunity both in individuals and in populations at scale.

The reported accuracy of antibody tests depends in part on the samples used in validation. We used a large and varied collection of 230 pre-pandemic samples collected from both children and adults in the UK and Europe across multiple years. Using these diverse cohorts, we established robust thresholds optimised to maximise specificity (~98%), which were maintained when evaluated in a second set of samples. Intriguingly, we observed increased background reactivity in adults compared to children across the 5 pre-pandemic cohorts tested for all assays. This finding contrasts with others who have reported higher cross-reactivity with serum antibody to seasonal HCoVs in younger populations (children and adolescents) than in adults[26], whilst others report no association with age[28]. These differences may reflect different trends in circulating viruses at the time of sample collection for each of the cohorts and/or differences between saliva and serum.

We observed significantly greater sensitivity for recent SARS-CoV-2 infection using assays for anti-spike compared to anti-RBD and N-protein, in line with other studies using serum and saliva[7,9,29]. The poorer performance of the RBD assays was surprising and contrasts with findings in serum where RBD can be used as a specific antigen for detection of SARS-CoV-2 infection, with responses mirroring those for spike[1]. This poor discriminatory performance was particularly notable for the RBD IgA assay. The cause of this is unclear, but others have reported similar findings with saliva samples[30]. One possible explanation may be that the pH of saliva alters antigen conformation, promoting non-specific binding. We report lower test sensitivity compared to serological tests (50.6% cf. ~98%). This finding is perhaps expected given the intrinsic variation and lower antibody concentrations associated with mucosal samples[9,12,21]. Despite this, salivary samples offer a unique opportunity to measure both systemic and mucosal responses non-invasively, as well as directly to detect and quantify levels of respiratory virus[31]. Further work should consider alternative testing platforms that may provide improved test accuracy over ELISA[32].

In households undergoing SARS-CoV-2 infection, salivo-conversion was observed as soon as 4 days post infection (spike and N-protein IgA). Notably, most unvaccinated PCR -ve household members, who were all children, mounted detectable salivary IgA responses in the absence of IgG responses. Moreover, most (11/13) asymptomatic PCR +ve cases salivo-converted. Taken together, this suggests an early role for mucosal antibody in limiting infection[12,33] and supports observations of negative blood-based antibody testing for COVID-19 in children[34]. Salivary antibody offers potential for enhanced surveillance in settings where PCR testing is limited[12,26,35] and serological testing of children is hampered.

This study has highlighted several considerations for future deployment of salivary antibody assays to SARS-CoV-2 and other infections. We observed variation in the type of salivary antibody responses and dynamics both within and between individuals, and both in magnitude and duration. Given the high reproducibility of the assays and control over sample collection methods, it is likely at least some of this reflects intrinsic variability in saliva as a biological sample: there are intra- and inter-individual differences in salivary flow rate, hydration state and gingival health[11]. Others have suggested to control for this by normalising to total immunoglobulin[6,9,36,37], but this could be subject to the same inconsistencies, so that normalisation could amplify errors and/or mask specific responses[38]. Expressing concentration of antibody as a normalised OD (a ratio to a serum standard) is a simple expression that minimises intrinsic assay variation and laboratory workload for high-throughput surveillance. Subsequent interpolation and reporting in international binding antibody units/ml (BAU/ml) would allow for cross-laboratory comparisons and assay standardisation[39].

Current gold-standard methods for detecting current infection (PCR) and past infection (serology) require invasive samples such as nasal/throat swabs or bloods. Saliva samples obtained by spitting are non-invasive and can be easily provided using non-specialist equipment, even from younger children. Moreover, we demonstrate that saliva samples are robust to sample handling and processing (heat inactivation and freeze-thawing) meaning that pre-treatment following collection isn't required and can be performed at a later date: this has implications for immediate testing but also provides assurance for retrospectively analysing existing collections of samples with similar test platforms. Finally, using wild-type antigen[1] we demonstrated applicability of assays to recent outbreaks when variants of concern (Delta and Omicron) were dominant. This has implications for future assay design, suggesting that, to date, wild-type antigen is robust in the face of new variants.

Our study has several limitations. We did not evaluate analytical specificity to other seasonal human coronaviruses (HCoVs) nor other respiratory viruses using our large pre-pandemic collection, where presence of antibodies to other confirmed coronaviruses may account for some false-positive results[29]. However, anti-spike salivary antibody responses have been demonstrated to be highly specific by others[9,40]. When we performed the test accuracy aspects of this study, we were unable to obtain 200 samples from recovered PCR-confirmed individuals as per MHRA guidelines, so estimates of test sensitivity are uncertain[20]. Field-testing was conducted during periods of high vaccine coverage among adults and as a result households did not include unvaccinated adults or vaccinated children. Furthermore, as generation of mucosal antibody responses were seen to occur at least 4 days from confirmed infection (salivo-conversion), individuals infected in the final weeks of household testing may have salivo-converted outside of the observation period. Although this was not observed, it should remain a consideration for future studies investigating kinetics of mucosal antibody responses. Finally, deployment of the best performing anti-spike assays for salivo-surveillance in vaccinated populations presents challenges: these assays cannot distinguish infected from vaccinated individuals, while anti-N-protein salivo-conversion appears to occur infrequently in infected individuals. Nonetheless, we did observe clear increases in salivo-positivity following infection in vaccinated individuals (in the household study), offering a potential means to identify periods of transmission when deployed in a mixed population.

Our findings emphasise the need for further work on understanding factors associated with SARS-CoV-2 mucosal antibody profiles and the heterogeneity in responses observed. Ongoing monitoring of mucosal antibody responses is essential for understanding transmission of SARS-CoV-2 and informing vaccination strategies, especially if future candidate vaccines are to be administered intranasally[37,41,42]. The rapidly increasing complexity of COVID-19 epidemiology globally requires tools to guide difficult policy decisions, especially for vaccination[43], and for countries with limited data on population immunity. Antibody assays should continue to be evaluated in the populations they are deployed to, particularly in landscapes with high numbers of infections and varying levels of pre-existing immunity. Multiplex salivary immunoassays could achieve the best diagnostic discrimination and offer additional insight for epidemiological inference[44,45]. Compared to single-plex systems, multiplexing costs are typically higher (e.g., increased antigen product and specialised systems). But, if affordable and high throughput, this approach can offer a means for long-term salivo-surveillance in hard-to-reach communities. In summary, we present methods for detecting salivary antibody and demonstrate feasibility of approach for large scale salivo-epidemiology. This approach for monitoring infection and immunity, using saliva as an easily obtainable non-invasive sample, that can be assayed simply and affordably, has the potential to gather data in places where information is scarce.

## Data availability

Source and aggregate data underlying Figs. 2–5 are available in Supplementary Data files 1 to 4, respectively. The datasets generated during the current study relating to assay development and evaluation are restricted in access but available on request from the University of Bristol data repository, data.bris, via https://doi.org/10.5523/bris.1urnu8sfg88322u2a9qh004zh6[46]. For data relating to field-testing in household outbreaks, consent was not obtained for all individuals to allow onward sharing of individual-level data to those external to the study. As such, raw data is not available for the full cohort. However, data from those who did consent to onward sharing will be available for request at the University of Bristol data repository, data.bris once data processing has been completed by searching for the CoMMinS study.

## Code availability

Machine learning analysis is available as a Jupyter notebook at https://github.com/Bristol-UNCOVER/Saliva_data_ML_analysis/blob/main/Saliva_dataset_analysis.ipynb. Archived source code at time of publication: https://doi.org/10.5281/zenodo.724998947

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

## Acknowledgements

The authors thank the Bristol BioBank for enabling access to pre-pandemic samples, the researchers involved in originally collecting these samples, and the study participants who agreed to donate them for future research. We acknowledge Helen Thompson for supporting organisation of COVID-19 Bristol BioBank clinics and Drs Jane Metz, Khuen Foong Ng, Charlie Plumptre and Jill King for supporting sample collection. We acknowledge Drs Philippa Lait and Chris Helps for support in establishing and evaluating SARS-CoV-2 RT-qPCR testing. We thank Bristol UNCOVER group for supporting the discussion of method development and interpretation of results. We thank Tessa Alexander for support in preparing data for publication. We acknowledge funding support from The University of Bristol and the Elizabeth Blackwell Institute supported by Bristol Alumni and Friends for equipment and reagents to conduct assay development and test accuracy studies. Deployment of assays to household outbreaks in the CoMMinS study was supported by the MRC [MR/V028545/1]. AT is supported by the Wellcome Trust (217509/Z/19/Z) and UKRI through the JUNIPER consortium MR/V038613/1 and CoMMinS study MR/V028545/1. E.B.P. and H.E.J. were partly supported by the NIHR Health Protection Research Unit (HPRU) in Behavioural Science and Evaluation. The views expressed are those of the author(s) and not necessarily those of the NHS, the NIHR or the Department of Health. The NIHR had no role in writing the manuscript or the decision to publish it. E.B.P. is funded via the JUNIPER Consortium (MRC grant no. MR/V038613/1) and MRC grant no. MC/PC/19067. NJT is a Wellcome Trust Investigator (202802/Z/16/Z), is the PI of the Avon Longitudinal Study of Parents and Children (MRC & WT 217065/Z/19/Z), is supported by the University of Bristol NIHR Biomedical Research Centre (BRC-1215-2001), the MRC Integrative Epidemiology Unit (MC_UU_00011/1) and works within the CRUK Integrative Cancer Epidemiology Programme (C18281/A29019).

## Author contributions

A.T., E.B.P., A.F., M.B. and A.H. conceived the study. A.T., E.O., H.B., J.S., B.H., U.O., A.H., H.A. and D.S. performed ELISA experiments. K.G., N.B., K.V., F.R., A.To. and I.B. produced antigen. A.T., K.S., A.H., H.B., A.L., E.O., H.J., M.B. and E.B.P. carried out computational analysis. B.M.A., K.D. and A.D. performed PCR experiments. A.T., E.O., J.S., B.H., J.O., B.M.A., F.R., R.B., L.C., G.G.L., H.D., A.G. and the CoMMinS Study Team collected and managed samples. R.B., L.C., N.G., G.G.L., H.D., A.G. and the CoMMinS Study Team collected clinical/demographic data. E.B.P., C.R., A.To., D.W., I.B., A.D. and K.Gi. contributed resources. E.B.P., M.B., A.F. and A.H. supervised. A.T., E.O., H.B., K.S., H.J., M.B., E.B.P. and A.H. prepared the original draft. All authors interpreted data, reviewed and edited the manuscript.

## Competing interests

The authors declare the following competing interests: AF is a member of the Joint Committee on Vaccination and Immunisation, the UK national immunisation technical advisory group and is chair of the WHO European regional technical advisory group of experts (ETAGE) on immunisation and *ex officio* a member of the WHO SAGE working group on COVID vaccines. He is investigator on studies and trials funded by Pfizer, Sanofi, Valneva, the Gates Foundation and the UK government.

**Additional information**

[1]Population Health Sciences, Bristol Medical School, University of Bristol, Bristol, UK. [2]Bristol Vaccine Centre, School of Cellular and Molecular Medicine, University of Bristol, Bristol, UK. [3]School of Biochemistry, University of Bristol, Bristol, UK. [4]BrisSynBio, University of Bristol, Bristol, UK. [5]Imophoron Ltd, Science Creates, Old Market, Midland Road, Bristol, UK. [6]School of Chemistry, University of Bristol, Bristol, UK. [7]Translational Health Sciences, Bristol Medical School, University of Bristol, Bristol, UK. [8]NIHR Blood and Transplant Research Unit in Red Cell Products, University of Bristol, Bristol, UK. [9]Bristol Veterinary School, University of Bristol, Bristol, UK. [10]Bristol Vaccine Centre, Population Health Sciences, University of Bristol, Bristol, UK. [11]School of Cellular and Molecular Medicine, University of Bristol, Bristol, UK. [12]Bristol Bioresource Laboratories, Population Health Sciences, Bristol Medical School, University of Bristol, Bristol, UK. [13]Bristol Royal Hospital for Children, University Hospitals Bristol and Weston NHS Foundation Trust, Upper Maudlin Street, Bristol BS2 8BJ, UK. [14]NIHR Bristol Biomedical Research Centre, University of Bristol, Bristol, UK. [15]Hospital Pediátrico, Centro Hospitalar e Universitário de Coimbra, Coimbra, Portugal. [16]Faculdade de Medicina, Universidade de Coimbra, Coimbra, Portugal. [17]MRC Integrative Epidemiology Unit, Population Health Sciences, Bristol Medical School, Bristol, UK. [18]Max Planck Bristol Centre for Minimal Biology, University of Bristol, Bristol, UK. [19]Paediatric Immunology & Infectious Diseases, Bristol Royal Hospital for Children, Bristol, UK. [23]These authors contributed equally: Adam Finn, Alice Halliday. *A list of authors and their affiliations appears at the end of the paper. ✉email: amyc.thomas@bristol.ac.uk

## the CoMMinS Study Team

Hanin Alamir[20], Holly E. Baum[20], Anu Goenka[20], Alice Halliday[20], Ben Hitchings[20], Elizabeth Oliver[20], Debbie Shattock[20], Joyce Smith[20], Amy C. Thomas[20], David Adegbite[21], Rupert Antico[21], Jamie Atkins[21], Edward Baxter[21], Lindsay Bishop[21], Adam Boon[21], Emma Bridgeman[21], Lucy Collingwood[21], Catherine Derrick[21], Leah Fleming[21], Ricardo Garcia Garcia[21], Guillaume Gonnage Liveria[21], Niall Grace[21], Lucy Grimwood[21], Jane Kinney[21], Rafaella Myrtou[21], Alice O'Rouke[21], Jenny Oliver[21], Chloe Payne[21], Rhian Pennie[21], Millie Powell[21], Laura Ratero Garcia[21], Aoife Storer-Martin[21], John Summerhill[21], Amy Taylor[21], Zoe Taylor[21], Helen Thompson[21], Samantha Thomson-Hill[21], Louis Underwood[21], Gabriella Valentine[21], Stefania Vergnano[21], Amelia Way[21], Maddie White[21], Arthur Williams[21], David Allen[22], Josh Anderson[22], Mariella Ardeshir[22], Michael Booth[22], Charles Butler[22], Monika Chaulagain[22], Alex Darling[22], Nicholas Dayrell-Armes[22], Kaltun Duale[22], Malak Eghleilib[22], Chloe Farren[22], Danny Freestone[22], Jason Harkness[22], William Healy[22], Milo Jeenes Flanagan[22], Maria Khalique[22], Nadine King[22], Anna Koi[22], Maia Lyall[22], Begonia Morales-Aza[22], Maria Pozo[22], Ainhoa Rodriguez Pereira[22], Jessica Rosa[22], Louise Setter[22], Liam Thomas[22], Dylan Thomas[22] & Jonathan Vowles[22]

[20]Immunology Lab Team, Bristol Vaccine Centre, School of Cellular and Molecular Medicine, University of Bristol, Bristol, UK. [21]School Surveillance Team, Bristol Vaccine Centre, School of Cellular and Molecular Medicine, University of Bristol, Bristol, UK. [22]Microbiology and Molecular Lab Team, Bristol Vaccine Centre, School of Cellular and Molecular Medicine, University of Bristol, Bristol, UK.

