## [Peer Review File · Communications Medicine]

Reviewers' comments:

Reviewer #1 (Remarks to the Author):

The authors describe the validation of salivary immunoassays including six ELISAs targeting IgG and IgA to spike, nucleocapsid and RBD antigens as a measure of past SARS-CoV-2 infection. The assays were assessed using accuracy studies (pre-pandemic samples and samples from individuals with PCR-confirmed infection). In addition, serum and saliva antibody responses were compared, and the assays were evaluated in the field in household outbreak studies.

General comments:

The authors present thorough and detailed validation results for a method that has practical advantages over the use of serum for detection previous SARS-CoV-2 infection, and have employed a variety of methods using a range of samples to evaluate the performance of the assays. While the study presents important data, the manuscript is very long and detailed (currently 8571 words from introduction to end of methods). The authors should consider reducing the content of the manuscript to the most important findings, and further details, where needed, can be moved to supplementary material. While this is not essential, I think it would significantly improve the readability of the manuscript.

Methods:

- Line 456 – Figure S9 does not show the SOP for saliva collection as indicated
- Lines 534 – reference number for Halliday et al. should be added
- Lines 693-694: Please can the authors clarify why PCR was performed on the same saliva sample as tested for antibodies – PCR would detect acute infection and antibodies would detect previous infection and would take approximately 10-14 days for the antibody response to develop post-infection. Using the same sample would likely limit the sensitivity of one of these assays depending on the time of sample collection post infection.

Results:

- Lines 81 and 82 and Figure S2 – they authors state that the different heat inactivation times did not affect the ELISA signal. From Figure S2, it appears that the OD was reduced for some samples at 30 min. Was statistical analysis done to compare and confirm no statistical difference?
- The assay with the highest sensitivity was spike IgG, however this sensitivity was only 50.6%, which is low for a diagnostic assay. Assay performance was improved when combining readings from multiple assays (highest when combining N, RBD and spike IgG assays) which would require multiple assays being conducted for each sample to achieve acceptable sensitivity, and thereby increase the time and cost, which may negate the benefit of using saliva samples over serum samples.
- Figure 4 – a number of pre-pandemic individuals displayed saliva IgA and IgG responses. Can the authors please explain this, and whether it is a specificity issue with the assays? How would this affect the interpretation of results diagnosing previous infection?

- Lines 242 – please clarify why there are 20 households but only 19 index cases. In addition, if 11 contacts were positive on day 0, these would be considered co-primary cases and not contacts. Please clarify.
- Table 3 – consider moving this table to the supplementary material as the characteristics of these household participants are not essential to the evaluation of the performance of the assays.
- Table 4 – Antibody responses were detected in a significant proportion of PCR negative contacts (71% spike IgG or IgA). This is concerning as it may indicate a large number of false positive results using these assays. In addition (Table 5), about a quarter of PCR negative individuals sero-converted during the follow-up period.
- Household cohorts – in this cohort of individuals, can the time from PCR positivity to detection of IgG and IgA salivary responses be determined? Does this differ from the known time to detection of serum antibodies?

Discussion:

- As specified in the results section comments, there are a number of limitations to these saliva-based immunoassays including low sensitivity when used in isolation, and therefore the need to use >1 assay to increase sensitivity, as well as the antibody responses detected in pre-pandemic samples and PCR negative individuals (false positive results). This should be made more clear in the study conclusions and abstract.

Reviewer #2 (Remarks to the Author):

Manuscript is well written and content is relevant for current research into SARS-CoV-2 diagnostics. Authors should consider the following minor comments and suggestions:

Lines 189-193: no values provided for PPV and NPV. Consider including to put interpretation into context

Lines 199 – 200: some specificities listed as actual values while some are ranges Be consistent in mentioning

Line 347: Discussion should mention limitations of current “gold-standard” in use for diagnosis of SARS CoV2, i.e. why would using saliva samples be more user friendly

Line 420: Limitations related to field study should mention exclusion of unvaccinated adults and vaccinated children.

Line 491: authors should specify if saliva was treated after collection and pre-transportation or pre-treated prior to freezing at -70.

Line 557: Sentence should be: “Final assay conditions were as follows: Antigens.....”

Line 600: Section on “Salivary ELISA development” should move before actual methods provided, i.e. to line 552.

Line 704: It is not clear whether COVID-19 positive participants were selected based on a PCR positive result obtained using the “gold-standard” diagnostic method employed by health-care facilities or an in house developed method on saliva. If the latter, why was the “SARS CoV2 infection by RT-qPCR on saliva” done using an in house developed method instead of the “gold standard” diagnostic used by health care facilities?

Line 734: Data and Statistical analysis. Authors should consider performing a method comparison using the Passing-Bablok regression for comparing the gold standard for

determining COVID-19 infection (PCR) to their established method of saliva antibody detection.

Evaluation and deployment of isotype-specific salivary antibody assays for detecting previous SARS-CoV-2 infection in children and adults

We thank the reviewers for taking the time to review the manuscript and for their constructive comments. Please see our responses in blue below.

Referee expertise:

Referee #1: diagnostics of respiratory pathogens, including SARS-CoV-2

Referee #2: diagnostics, immunoassays

Reviewers' comments:

Reviewer #1 (Remarks to the Author):

General comments:

The authors present thorough and detailed validation results for a method that has practical advantages over the use of serum for detection previous SARS-CoV-2 infection, and have employed a variety of methods using a range of samples to evaluate the performance of the assays. While the study presents important data, the manuscript is very long and detailed (currently 8571 words from introduction to end of methods). The authors should consider reducing the content of the manuscript to the most important findings, and further details, where needed, can be moved to supplementary material. While this is not essential, I think it would significantly improve the readability of the manuscript.

Thank you for these suggestions. We have made substantial changes to the manuscript to reduce content where possible, these edits have focused on refinement mostly within the results section and movement of certain methodological details to the supplementary material. Specifically, edits include:

- Movement of methods detailing pre-pandemic sample collection to the supplementary information.
- Movement of PCR methods to the supplementary information.
- Movement of details on QC material generation to the supplementary information.
- Movement of details on production of protein for ELISA.
- Movement of Table 3 to the supplement as suggested below.

The current length is 6,381 words, 2,781 of these detail methods.

Methods:

- Line 456 – Figure S9 does not show the SOP for saliva collection as indicated
Thank you for this observation. This is an error - inclusion of the SOP was omitted. Instead, full details of the protocol are listed in the methods text. We have now removed this sentence.
- Lines 534 – reference number for Halliday et al. should be added
Thank you, details of this reference are now included.
- Lines 693-694: Please can the authors clarify why PCR was performed on the same saliva sample as tested for antibodies – PCR would detect acute infection and antibodies would

detect previous infection and would take approximately 10-14 days for the antibody response to develop post-infection. Using the same sample would likely limit the sensitivity of one of these assays depending on the time of sample collection post infection.

Systemic antibody responses post primary exposure to SARS-CoV-2 infection and/or vaccination have been well described in the literature, whereas mucosal antibody responses, in the context of varying prior infection and/or vaccination status, are relatively unclear. We therefore sought to combine matched PCR and antibody testing on longitudinally collected saliva samples in the context of recent transmission, to gain a better understanding of these relationships.

Results:

- Lines 81 and 82 and Figure S2 – they authors state that the different heat inactivation times did not affect the ELISA signal. From Figure S2, it appears that the OD was reduced for some samples at 30 min. Was statistical analysis done to compare and confirm no statistical difference?

Thank you for this suggestion. We agree that the OD appears reduced for some samples at 30 minutes, but for other samples, the OD appears increased. We have now performed statistics using the Friedman's test and can confirm that OD is not statistically different between treatments. These results are reflected in an updated Figure S2 legend.

- The assay with the highest sensitivity was spike IgG, however this sensitivity was only 50.6%, which is low for a diagnostic assay. Assay performance was improved when combining readings from multiple assays (highest when combining N, RBD and spike IgG assays) which would require multiple assays being conducted for each sample to achieve acceptable sensitivity, and thereby increase the time and cost, which may negate the benefit of using saliva samples over serum samples.

We agree that a diagnostic sensitivity of 50.6% is low and this is acknowledged in the abstract and discussion (lines 426-227 and 469-470). To overcome this reduction in sensitivity, we propose use of multiplex assays i.e., a single assay that measures responses to multiple antigens in the same sample (last discussion para). Multiplex assays shouldn't increase labour time, since it's still 1 assay/sample, but we agree they can increase costs compared to single-plex systems. We have enhanced this section of the discussion through acknowledging the associated trade-offs with the multiplex approach as follows:

Lines 556-560: "Multiplex salivary immunoassays could achieve the best diagnostic discrimination and offer additional insight for epidemiological inference^{37, 38}. Compared to single-plex systems, multiplexing costs are typically higher (e.g., increased antigen product and specialised systems). But, if affordable and high throughput, this approach can offer a means for long-term salivo-surveillance in hard-to-reach communities."

- Figure 4 – a number of pre-pandemic individuals displayed saliva IgA and IgG responses. Can the authors please explain this, and whether it is a specificity issue with the assays? How would this affect the interpretation of results diagnosing previous infection?

Thank you for this observation. Thresholds were optimised and validated to achieve >98% specificity, interpretation of results for diagnosing recent infection should not be affected, since these thresholds are robust to false positives.

Specificity is best evaluated in Figure 2 using the proposed and final thresholds for reference – specificity and sensitivity was poor RBD (in agreement with Figure 4). As a result, we did not take forward the RBD assays for full evaluation. Reasons for the poor RBD specificity are hypothesised in the discussion (3rd para).

- Lines 242 – please clarify why there are 20 households but only 19 index cases. In addition, if 11 contacts were positive on day 0, these would be considered co-primary cases and not contacts. Please clarify.

Thank you for this observation. We have clarified that the 20th index case did not provide sufficient saliva samples for antibody analysis and was subsequently excluded from analysis. Lines 343-344: “Note one index case did not provide sufficient saliva for analysis.”

We agree that the 11 contacts with prevalent infection (PCR +ve at Day 0) can represent co-primary cases. However, the definition of an index case in this setting does not represent the first household member to become infected, instead it refers to the first household member to self-identify with confirmed SARS-CoV-2 infection (thereby triggering enrolment to the household study). We have clarified this in the methods section ‘Study participants’ with the additional sentence (lines 585-586):

“The individual that self-identified as SARS-CoV-2 positive is subsequently termed the index case.”

- Table 3 – consider moving this table to the supplementary material as the characteristics of these household participants are not essential to the evaluation of the performance of the assays.

In line with general comments above to reduce content, Table 3 has been moved to the supplementary material and has become Table S11.

- Table 4 – Antibody responses were detected in a significant proportion of PCR negative contacts (71% spike IgG or IgA). This is concerning as it may indicate a large number of false positive results using these assays. In addition (Table 5), about a quarter of PCR negative individuals sero-converted during the follow-up period.

Given the assays are optimised and validated to achieve high specificity ($\geq 98\%$), false positives are highly unlikely to occur ($\leq 2\%$).

Antibody responses in PCR -ve’s likely represent pre-existing immunity – either historical infection prior to study enrolment and/or vaccination:

- PCR -ve antibody positive individuals on Day 0 may have cleared infection and could be ‘true index’ cases (see lines 373-375)
- Anti-spike IgA or IgG was detected in 18/20 vaccinated PCR -ve contacts, reflecting vaccine induced (and possibly boosted) responses (Table S12).

Alternatively, considering the sampling context, contacts have known exposure to some level of virus during a household outbreak. Therefore, we hypothesise that detection of salivary antibody in absence of PCR confirmed infection may represent an early role for mucosal antibody in limiting infection (possibly below the LoD of PCR). This is the small group of PCR-ve contacts that salivo-convert (Table 4).

In response to these comments please note Table 4 has now become Table 3 and Table 5 has become Table 4.

- Household cohorts – in this cohort of individuals, can the time from PCR positivity to detection of IgG and IgA salivary responses be determined? Does this differ from the known time to detection of serum antibodies?

Thanks for this interesting point – yes, it is possible to calculate for some household members with incident infection (i.e., become PCR +ve during the study and subsequently salivo-convert), although numbers are small. We remark on this in the discussion having observed

salivo-conversion to occur as soon as 4 days post infection. We have elaborated on this point to give details of antigen/isotype (spike and N-protein IgA). Kinetics can also be visualised in Figure S10.

Lines 484-485: “In households undergoing SARS-CoV-2 infection, salivo-conversion was observed as soon as 4 days post infection (spike and N-protein IgA).”

Discussion:

- As specified in the results section comments, there are a number of limitations to these saliva-based immunoassays including low sensitivity when used in isolation, and therefore the need to use >1 assay to increase sensitivity, as well as the antibody responses detected in pre-pandemic samples and PCR negative individuals (false positive results). This should be made more clear in the study conclusions and abstract.

We have emphasised the low sensitivity of assays in the conclusions (first para of discussion) and abstract.

Lines 426-427: “However, sensitivity is considerably reduced compared to conventional serum-based tests.”

Lines 9-10: “However, sensitivity was low for the best performing assay (spike IgG: 50.6%, 39.8-61.4%).”

As above, the PCR -ve individuals with antibody above the threshold for positivity are highly unlikely to represent false positives (considering data from both pre-pandemic donors and household members).

Reviewer #2 (Remarks to the Author):

Manuscript is well written and content is relevant for current research into SARS-CoV-2 diagnostics. Authors should consider the following minor comments and suggestions:

Lines 189-193: no values provided for PPV and NPV. Consider including to put interpretation into context

Thank you for this suggestion. We have included values of PPV and NPV as suggested. This now reads:

Lines: 189-192: “Positive predictive value at 5% prevalence was higher (fewer false positives) in the spike IgA (30.6-45.7%), IgG (66-82.4%) and N-protein IgG (30.1-35.1%) assays compared to N-protein IgA (9.8-11.5%). NPV was lowest (increased false negatives) for N-protein IgA (95.1-95.3%). PPV and NPV were robust up to 40% prevalence (Figure S7).”

Lines 199 – 200: some specificities listed as actual values while some are ranges Be consistent in mentioning

Thank you for this observation, we have amended to be consistent as follows:

Lines 179-182: “Considering false positivity by age, specificity was 100% in younger age groups (0-19 years) for spike IgA, IgG and N-protein IgG (10-19 years) assays. Specificity was lower for N-protein IgA (0-9 years: 98.2%; 10-19 years: 88.9%) and N-protein IgG (0-9 years: 98.2%). The lowest observed specificity was for spike IgG in 30-39 years (81.8%) (Table S7).”

Line 347: Discussion should mention limitations of current “gold-standard” in use for diagnosis of SARS CoV2, i.e. why would using saliva samples be more user friendly

Thank you for this suggestion. We have included limitations of the current gold-standard tests in the discussion:

Lines 507-514: “Current gold-standard methods for detecting current infection (PCR) and past infection (serology) require invasive samples such as nasal/throat swabs or bloods respectively. Saliva samples obtained by spitting are non-invasive and can be easily collected using non-specialist equipment, even from younger children.”

Line 420: Limitations related to field study should mention exclusion of unvaccinated adults and vaccinated children.

We have included this limitation in the discussion:

Lines 529-531: “Furthermore, field-testing was conducted during periods of high vaccine coverage among adults, resultingly households did not include unvaccinated adults or vaccinated children.”

Line 491: authors should specify if saliva was treated after collection and pre-transportation or pre-treated prior to freezing at -70.

Thank you for this observation, we have included further details as follows:

“Moreover, we demonstrate that saliva samples are robust to sample handling and processing (heat inactivation and freeze-thawing) meaning that pre-treatment following collection isn’t required and can be performed at a later date: this has implications for immediate testing but also provides assurance for retrospectively analysing existing collections of samples with similar test platforms.”

Line 557: Sentence should be: “Final assay conditions were as follows: Antigens.....”

This has been corrected, thank you.

Line 600: Section on “Salivary ELISA development” should move before actual methods provided, i.e. to line 552.

This section on ‘Salivary ELISA development’ has been moved before the section ‘Conduct of immunoassays’ as suggested.

Line 704: It is not clear whether COVID-19 positive participants were selected based on a PCR positive result obtained using the “gold-standard” diagnostic method employed by health-care facilities or an in house developed method on saliva. If the latter, why was the “SARS CoV2 infection by RT-qPCR on saliva” done using an in house developed method instead of the “gold standard” diagnostic used by health care facilities?

Field-testing antibody assays was conducted using samples collected during household outbreaks. Households were enrolled as part of a larger study (CoMMinS) to investigate transmission in this setting (details given in ‘Methods’, ‘Study participants’). PCR testing of saliva in CoMMinS was conducted using an in-house PCR assay - if individuals were found to be positive, they were advised to get a standard government test.

Performance of the in-house PCR was evaluated using 60 positive and 59 negative samples in collaboration with diagnostic personnel from health-care facilities in our region (UKHSA): 58/60 positive samples were found to be positive, no negative samples were positive.

Line 734: Data and Statistical analysis. Authors should consider performing a method comparison using the Passing-Bablok regression for comparing the gold standard for determining COVID-19 infection (PCR) to their established method of saliva antibody detection.

Thank you for raising our attention to Passing-Bablok regression. We are unable to perform this analysis as the procedure uses continuous measures – we do not have Ct values for

known positives in the combined threshold and validation sample set. Moreover, further investigation of the approach suggests it's better suited to comparison of the same type of measure generated by different tests i.e., comparison of two immunoassays (e.g., Perkmann et al. 2022, [https:// doi.org/10.1128/spectrum.01402-21](https://doi.org/10.1128/spectrum.01402-21)).

Reviewers' comments:

Reviewer #1 (Remarks to the Author):

Overall, I am satisfied with the responses from the authors, and the revised manuscript.

There is one comment that I think has not been fully addressed, and I suggest that a limitation is included in this regard.

Re this comment: "• Lines 693-694: Please can the authors clarify why PCR was performed on the same saliva sample as tested for antibodies – PCR would detect acute infection and antibodies would detect previous infection and would take approximately 10-14 days for the antibody response to develop post-infection. Using the same sample would likely limit the sensitivity of one of these assays depending on the time of sample collection post infection.

Systemic antibody responses post primary exposure to SARS-CoV-2 infection and/or vaccination have been well described in the literature, whereas mucosal antibody responses, in the context of varying prior infection and/or vaccination status, are relatively unclear. We therefore sought to combine matched PCR and antibody testing on longitudinally collected saliva samples in the context of recent transmission, to gain a better understanding of these relationships."

Thank you for this explanation, however this does not account for the fact that antibody responses, even those that are mucosal, would take time to develop and may not be present at the time that the PCR sample was collected during acute infection. This should be mentioned in the limitations. As noted further down in support of this, individuals seroconverted at a minimum of 4 days post-infection.

Reviewer #2 (Remarks to the Author):

Thank you for the revised article. The inclusions of a supplementary information section gives the article a more professional feel.

Responses to comments appreciated.

Few additional comment:

Lines 189-193: Although PPV and NPV have added, ranges have been inserted. Please specify what the range indicates. PPV and NPV should be single values

Line 496: Is 39404040 a reference?

Line 612: is 4243 a reference?

Evaluation and deployment of isotype-specific salivary antibody assays for detecting previous SARS-CoV-2 infection in children and adults

We thank the reviewers for taking the time to consider our responses and for their constructive comments. Please see our responses in blue below.

Reviewer #1 (Remarks to the Author):

Overall, I am satisfied with the responses from the authors, and the revised manuscript.

There is one comment that I think has not been fully addressed, and I suggest that a limitation is included in this regard.

Re this comment: "• Lines 693-694: Please can the authors clarify why PCR was performed on the same saliva sample as tested for antibodies – PCR would detect acute infection and antibodies would detect previous infection and would take approximately 10-14 days for the antibody response to develop post-infection. Using the same sample would likely limit the sensitivity of one of these assays depending on the time of sample collection post infection. Systemic antibody responses post primary exposure to SARS-CoV-2 infection and/or vaccination have been well described in the literature, whereas mucosal antibody responses, in the context of varying prior infection and/or vaccination status, are relatively unclear. We therefore sought to combine matched PCR and antibody testing on longitudinally collected saliva samples in the context of recent transmission, to gain a better understanding of these relationships."

Thank you for this explanation, however this does not account for the fact that antibody responses, even those that are mucosal, would take time to develop and may not be present at the time that the PCR sample was collected during acute infection. This should be mentioned in the limitations. As noted further down in support of this, individuals seroconverted at a minimum of 4 days post-infection.

Thank you for this suggestion. We agree that by not accounting for the time taken for individuals to develop mucosal antibody, we may have missed observations of salivo-conversion during acute infection, i.e., should individuals have become infected in the final weeks of conducting household testing and salivo-converted outside the observation period. Although this was not observed for individuals with evidence of PCR confirmed infection in weeks 3 and 4, we have highlighted this as a potential limitation as follows:

Lines 380-384: "Furthermore, as generation of mucosal antibody responses were seen to occur at least 4 days from confirmed infection (salivo-conversion), individuals infected in the final weeks of household testing may have salivo-converted outside the observation period. Although this was not observed, it should remain a consideration for future studies investigating kinetics of mucosal antibody responses."

Reviewer #2 (Remarks to the Author):

Thank you for the revised article. The inclusions of a supplementary information section gives the article a more professional feel. Responses to comments appreciated.

Few additional comment:

Lines 189-193: Although PPV and NPV have added, ranges have been inserted. Please specify what the range indicates. PPV and NPV should be single values

Thank you for this suggestion, we have specified that these ranges relate to values of PPV and NPV at different threshold values as follows:

Lines 161-163: “Ranges for PPV and NPV indicate values at each threshold method (97th to 99th percentile and Youden’s index), estimates were robust up to 40% prevalence (Figure S7).”

Line 496: Is 39404040 a reference?

Line 612: is 4243 a reference?

Thank you for bringing these insertions to our attention, they were introduced in error and have been removed.

REVIEWERS' COMMENTS:

Reviewer #1 (Remarks to the Author):

Thank you for the response and for addressing the concern in the limitations, I have no further comments.

Reviewer #2 (Remarks to the Author):

No further comments